# High throughput error corrected Nanopore single cell transcriptome sequencing

Kevin Lebrigand [1✉], Virginie Magnone[1], Pascal Barbry [1✉] & Rainer Waldmann [1✉]

Droplet-based high throughput single cell sequencing techniques tremendously advanced our insight into cell-to-cell heterogeneity. However, those approaches only allow analysis of one extremity of the transcript after short read sequencing. In consequence, information on splicing and sequence heterogeneity is lost. To overcome this limitation, several approaches that use long-read sequencing were introduced recently. Yet, those techniques are limited by low sequencing depth and/or lacking or inaccurate assignment of unique molecular identifiers (UMIs), which are critical for elimination of PCR bias and artifacts. We introduce ScNaUmi-seq, an approach that combines the high throughput of Oxford Nanopore sequencing with an accurate cell barcode and UMI assignment strategy. UMI guided error correction allows to generate high accuracy full length sequence information with the 10x Genomics single cell isolation system at high sequencing depths. We analyzed transcript isoform diversity in embryonic mouse brain and show that ScNaUmi-seq allows defining splicing and SNVs (RNA editing) at a single cell level.

---

[1] Université Côte d'Azur, CNRS, Institut de Pharmacologie Moléculaire et Cellulaire, F06560 Sophia Antipolis, France. ✉email: lebrigand@ipmc.cnrs.fr; barbry@ipmc.cnrs.fr; rainer@ipmc.cnrs.fr

Single-cell RNA sequencing (scRNA-seq) is a key technique for the analysis of cell-to-cell heterogeneity and projects aiming at analyzing the transcriptome of all cells from complex organisms have been initiated (e.g., Human Cell Atlas[1], Tabula Muris[2]). While droplet-based high throughput scRNA-seq approaches (e.g., 10xGenomics Chromium) allow the analysis of thousands of cells, they only yield limited sequence information close to one extremity of the transcript after Illumina short-read sequencing. Information crucial for an in-depth understanding of cell-to-cell heterogeneity on splicing, chimeric transcripts and sequence diversity (SNPs, RNA editing, imprinting) is lacking.

Long-read sequencing can overcome this limitation. Several studies used Pacific Biosciences (PacBio) or Nanopore long-read sequencing to obtain full-length single-cell data with the 10x Genomics Chromium system. Gupta et al.[3] reported PacBio single-cell sequencing of 6627 cells. However, due to the low PacBio sequencing throughput, the sequencing depth was rather low (270 reads, 260 UMI, 129 genes per cell). Such a low depth limits analysis to highly expressed transcripts. Oxford Nanopore PromethION long-read sequencers generate 20 times more reads per flow cell than the PacBio Sequel II. The first Nanopore sequencing of 10x Genomics single-cell libraries was a targeted sequencing of antigen receptors in lymphocytes[4]. To assure correct barcode assignment, both studies used high accuracy Illumina sequencing data of the same libraries to guide cell barcode (cellBC) assignment to long reads. Volden et al.[5] recently presented a Nanopore sequencing approach with undisclosed barcode assignment accuracy, that does not require Illumina data for cellBC assignment.

Single-cell sequencing library preparation requires rather strong PCR amplification. Amplification bias and chimeric cDNA generated during PCR amplification are issues that can be both addressed by unique molecular identifiers (UMIs), short random sequence tags that are introduced during reverse transcription. UMIs allow grouping of reads that correspond to same RNA molecule and elimination of PCR artifacts. In consequence UMIs minimize the risk that PCR generated chimeric cDNAs are falsely annotated as novel transcripts. Furthermore, UMIs allow the generation of error-corrected consensus sequences for each RNA molecule. Obviously, accuracy of UMI assignment is crucial. Previous long-read single-cell sequencing approaches either did not use UMIs[4] or did not correct UMI sequencing errors[3,5]. They assigned every novel UMI read sequence to a novel UMI. However, with this strategy, the higher sequencing error rate of long-read sequencers causes serious issues. PacBio sequencers have a circular consensus error rate of about 1%. In consequence, about 10% of the 10 nt. UMI reads are expected to have at least one error. Fake UMIs are an even more serious issue with the higher Nanopore sequencing error rate of 5–8%. For 45% of the UMIs identified by Nanopore sequencing, Volden et al.[5] could not find a corresponding UMI in the Illumina dataset of the same sample. The authors suggest that at least one-third of the UMIs were misassigned with their strategy due to Nanopore sequencing errors.

We addressed those issues and designed a long-read single-cell sequencing approach that combines the high throughput of Nanopore sequencing with high accuracy cellBC and UMI assignment. Our approach, entitled **ScNaUmi-seq** (Single-cell Nanopore sequencing with UMIs), enables the analysis of splicing and sequence variation at the single-cell level with the 10x Genomics Chromium system. This is illustrated with data on alternative splicing and RNA editing in embryonic mouse brain.

## Results and discussion
### Assignment of cell barcodes and unique molecular identifiers to Nanopore reads. We prepared a 190 cell and a 951 cell E18

mouse brain library with the 10x Genomics Chromium system and generated $43 \times 10^6$ and $70 \times 10^6$ Illumina reads (Supplementary Fig. 1) as well as $32 \times 10^6$ and $322 \times 10^6$ Nanopore reads for the 190 and 951 cell replicates, respectively.

Since cellBCs and UMIs are located between a 3′ PCR priming site (adapter) and the polyA-tail of the cDNA (Supplementary Fig. 2b) we first searched for a >20 nt. sequence with at least 17 As within 100 nucleotides from the end of the read and then for the adapter downstream of the poly(A) tail. We identified both the poly(A) tail and the adapter in $57 \pm 11\%$ of the reads (Fig. 1b). This initial scan removed most of the low quality (QV < 10) and non genome-matched reads (Supplementary Fig. 3a–e).

To ensure highly accurate barcode assignment, we used a strategy where barcodes assignment is guided with Illumina data. We first extracted for each gene and genomic region (500 nt. windows) the barcodes detected in the Illumina short-read data. We then compared the cellBC sequence extracted from each genome aligned Nanopore read with the cell barcodes found in the Illumina data for the same gene or genomic region. Following this strategy, we assigned cellBCs to $68 \pm 4\%$ of Nanopore reads with identified poly(A) and adapter sequence (Fig. 1b; Supplementary Fig. 3a, c; see methods section for details).

The poly(A) and cellBC discovery rates of our approach are, despite the higher error rates of Nanopore reads, similar to those reported previously for PacBio sequencing of 10x Genomics libraries[3].

Molecular barcoding with UMIs facilitates elimination of PCR artifacts and sequencing error correction (Supplementary Fig. 4). Yet, high accuracy assignment of UMIs to long reads is challenging and was not reported as for yet. The principle reasons are: (i) long reads, in particular Nanopore reads, have a far higher error rate than Illumina reads. With a median Nanopore accuracy below 95%, more than half of the 10 nt. UMI reads are expected to have at least one error. (ii) Even at a high sequencing saturation, the majority of UMIs is covered by just a few reads (Fig. 1c). Clustering such unprecise UMI sequences with just a few reads per UMI is rather error prone. To avoid widespread UMI misassignment to Nanopore reads as reported by others[5], we designed a UMI assignment strategy that is guided by Illumina high accuracy sequencing data. After assignment of the cellBC to the Nanopore read, we compared the Nanopore UMI read sequence with the UMI sequences found for the same gene (or genomic region) and the same cell in the Illumina sequencing data (see methods section for details). This strategy drastically reduces the complexity of the UMI search set, which corresponds to the number of transcripts molecules captured for a given gene or genomic region in one cell. Using this strategy, we assigned UMIs to $76 \pm 3\%$ of the reads with identified cellBC (Fig. 1b).

We next examined the accuracy of our cellBC and UMI assignment strategy. We replaced either the cellBC or the UMI sequence in each Nanopore read by a random sequence and examined the number of cellBC and UMI assigned reads (see methods section for details). The accuracy ($100*(1 -\mathrm{assigned_{random}}/\mathrm{assigned})$) of both cellBC and UMI assignment were 99.8% and 97.4%, respectively (Fig. 1d). We also examined the precision of cellBC and UMI assignment with a second strategy where we compared the number of cellBC and UMI assigned reads obtained with the Illumina dataset of the same and of another unrelated 885 cell mouse brain sample. We found 140 times less cellBC and UMI assigned reads with the unrelated Illumina data, suggesting a combined cellBC/ UMI assignment accuracy of 99.3%. The accuracy is likely higher since cellBCs and UMIs were found at higher edit distances (ED) with the unrelated Illumina dataset (mean ED: cellBCs, 3.01; UMIs, 2.28; $n = 38,310$) than with the short-read

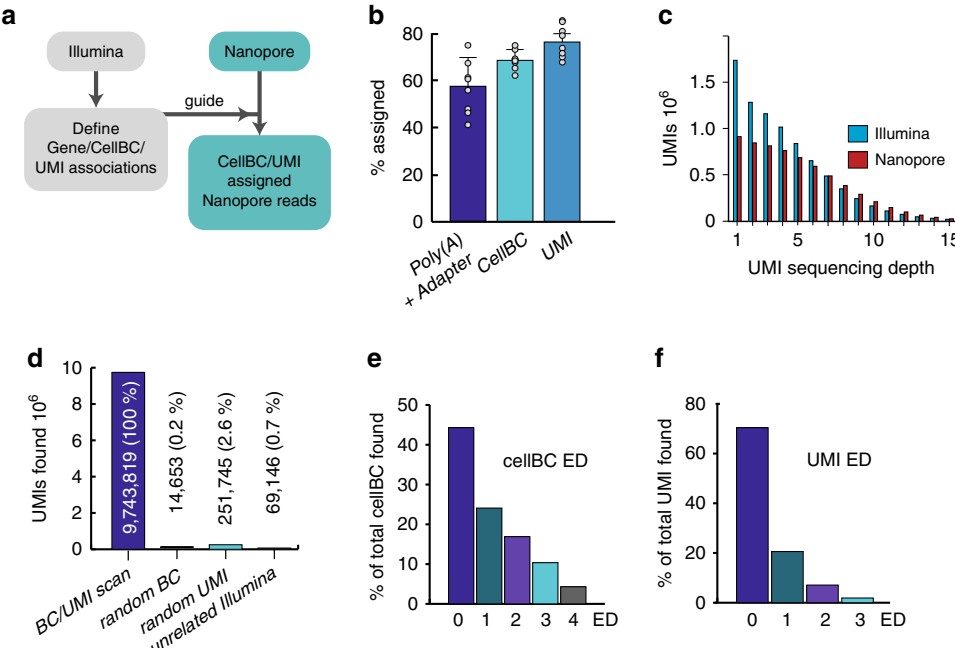

**Fig. 1 Efficiency and accuracy of cellBC and UMI assignment. a** CellBC and UMI assignment strategy (Detailed in Supplementary Fig. 2). **b** Efficiency of cellBC and UMI identification for eight PromethION sequencing runs ($n = 8$): both poly(A) and adapter, 57 ± 11% (SD) of total reads; cellBC, 68 ± 4% (SD) of reads with identified poly(A) and adapter; UMI, 76 ± 3% (SD) of reads with identified cellBC. Boxes and error bars indicate the means and standard deviations, respectively, for $n = 8$ flow cells. **c** UMI sequencing depth (reads/UMI) for the Illumina and Nanopore dataset. **d** Accuracy of cellBC and UMI assignment. Number of cellBC and UMI assigned reads before and after replacement of cellBC or UMIs with random sequences and after cellBC and UMI identification guided by an unrelated Illumina dataset ($24 \times 10^6$ reads of the 190 cell sample were scanned). **e, f** Edit distance distribution of the cellBC and UMI assignment.

data of the same sample (mean ED: cellBCs, 1.05; UMIs, 0.51; $n = 1{,}841{,}442$).

To evaluate whether the additional cycles of full-length PCR amplification (see "Methods" section) or Nanopore library preparation skew the long-read dataset, we compared the cellBC and UMI assigned Nanopore data to the Illumina data of the same sample. On average, 79% of the RNA molecules (UMIs) and 91% of the genes identified in each single cell after Illumina sequencing were also found in our Nanopore dataset (Fig. 2a, b). The cellBC and UMI assigned Nanopore reads (median 28,120 /cell) reflect a median of 2427 genes and 6047 UMIs per cell with a good correlation between Nanopore and Illumina gene counts (Fig. 2a; $r = 0.99$), UMI counts (Fig. 2b; $r = 0.99$) and gene expression for individual cells (Fig. 2c, mean $r = 0.90$). In consequence, our cellBC and UMI assigned Nanopore dataset represents well the transcriptome captured in the 10x Genomics workflow.

**Identification of transcript isoforms in full-length E18 mouse brain transcriptome**. We next analyzed the transcript isoforms in our long-read dataset. In total we found 33,002 Gencode vM18 annotated transcripts where all annotated exon-exon junctions were supported by at least one UMI. It was recently suggested that single cells tend to express dominantly one transcript isoform of a gene[6]. While we noticed this for certain genes (e.g., Pkm), we also found many instances of well-expressed genes showing expression of several isoforms in a single cell (Supplementary Fig. 5).

We also identified 4388 novel isoforms in our dataset (Supplementary Fig. 6, Supplementary Data 2, Supplementary Data 3). We required that novel transcript isoforms: (i) are backed by at least five UMIs; (ii) have all splice junctions confirmed in a mouse brain Illumina short-read dataset; (iii) have

a 5′ end within 50 nt. from CAGE-seq identified transcription start site; (iv) have a polyadenylation site within 50 nt. of the transcript 3′ end (see methods section and Supplementary Note for details). Globally, the novel isoforms were detected at far lower levels (Gencode: median 3795 UMIs/cell; novel: 60 UMIs/ cell) than Gencode isoforms suggesting that some of those novel isoforms might reflect a certain leakiness of the splicing or transcription machinery.

A t-SNE plot of the Illumina short-read gene expression data reveals typical cell types for E18 mouse brain (Fig. 2d). t-SNE projection of transcript isoform expression defined by Nanopore sequencing (Fig. 2e) yielded a similar clustering without revealing novel well-defined sub-clusters. Globally, the isoform-based clustering was more diffuse than the gene-based clustering. This is likely due to: (i) a split of the UMIs for a given gene in a cell between several isoforms (16,612 genes vs. 33,002 isoforms). This results in globally lower isoform counts and a higher dropout rate (cells with zero UMIs for a given isoform). (ii) Only 63.6% of the UMIs could be assigned to exactly one transcript isoform, resulting in a further reduction of isoform UMI counts (Illumina: median 7605 UMIs/cell; nanopore: median 6047 geneUMIs/cell, 3795 isoformUMIs/cell).

In two independent technical replicates, the 951 cell and the 190 cell datasets, the corresponding clusters correlated well in gene expression based on either Illumina or Nanopore data and in isoform expression deduced from Nanopore data (Supplementary Fig. 7).

We next searched for genes with a transcript isoform expression that differed between clusters. We noticed cell-type selective isoform usage for 76 genes and 174 differentially expressed isoforms (Supplementary Data 1, Supplementary Fig. 8b). For instance, Clathrin light chain A (Clta) (Fig. 3a–d) and Myosin Light Chain 6 (Myl6; Supplementary Fig. 9) undergo

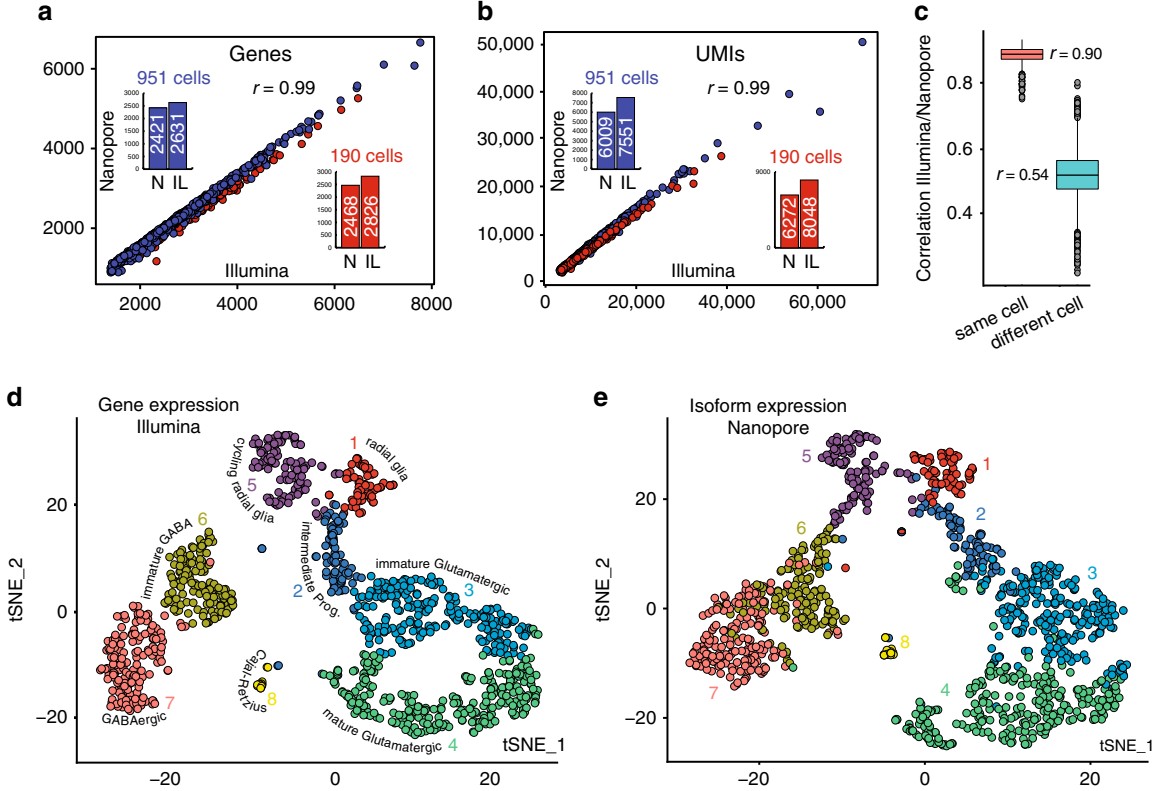

**Fig. 2 Comparison of Nanopore long read with short-read scRNA-seq.** Number of genes **a** and number of UMIs **b** detected for each cell of the 190 and 951 cell datasets after Illumina and Nanopore sequencing. Median genes **a** and UMIs **b** per cells for both datasets are shown in the histograms.
**c** Correlation of gene expression between Illumina and Nanopore sequencing data for each cell. Boxes represent the 25% quantile to 75% quantile range, upper and lower edges of notches are median $+/-$ 1.58 * IQR/sqrt(n) (IQR: inter quantile range, $n = 1121$). t-SNE plots of Illumina gene expression **d** and Nanopore isoform expression **e** for 1121 cells (integration of the 190 and 951 cell datasets). Colors for individual cells are the same in **d** and **e**. Clustering details are in Supplementary Fig. 8 and Supplementary Data 1.

a pronounced isoform switch during neuronal maturation. Myosin and Clathrin are involved in neuronal migration and axonal guidance[7,8] and in synaptic membrane recycling[9] and synaptic remodeling[10] associated with synaptic plasticity in mature neurons. The isoform switch of Clta and Myl6 might fine-tune both proteins for their respective roles at different developmental stages.

**ScNaUmi-seq can detect SNVs.** We next examined how single-cell Nanopore sequencing with UMIs (ScNaUmi-seq) performs in defining SNVs for the ionotropic glutamate receptor Gria2, a post-synaptic cation channel that is A->I edited at two sites, leading to a Q/R substitution within the pore that renders the channel $Ca^{2+}$ impermeable and a R/G substitution within the ligand-binding domain that results in accelerated recovery from activation[11,12]. UMIs allow addressing the principle weakness of Nanopore sequencing, the low accuracy. Generation of consensus sequences for single RNA molecules (UMIs) allows boosting the Nanopore sequencing accuracy from 93% to beyond 99% (Supplementary Fig. 4b, c) and identification of such sequence heterogeneity at a single-cell resolution (Fig. 4a, b). Analysis of error-corrected Gria2 consensus sequences confirmed previous findings[11] that Gria2 editing (Fig. 4c) is almost complete at the Q/R editing site (83.8%) and partial at the R/G editing site (20.8%) in E18 mouse brain. Long-read sequencing further revealed that editing of one site increases the probability that the other site is edited (Fig. 4c) and that editing of the R/G site increases during neuronal maturation (Fig. 4d) from 9.3% in neuronal progenitors to 26.7 and 26% in mature inhibitory and glutamatergic neurons

respectively. Thus, single-cell long-read sequencing both confirms and extends previous knowledge on Gria2 editing in the central nervous system.

Combining the high throughput of Nanopore sequencing with UMI guided error correction thus allows both high confidence definition of transcript isoforms and identification of sequence heterogeneity in single cells.

Accurate cellBC and in particular UMI assignment is crucial in single-cell sequencing. To achieve this, we opted for an approach that is guided by Illumina data. While this requires additional short-read sequencing, this strategy has multiple advantages. (i) We achieve a cellBC assignment accuracy that is, despite the higher error rate of Nanopore reads, comparable to that previously reported for PacBio reads[3]. (ii) This is the first approach that enables accurate UMI assignment to long reads. (iii) The additional short-read dataset of the same sample provides a quality control that allows detecting bias introduced by the additional amplification of full-length cDNA during Nanopore library preparation.

One principal advantage of our approach is the accurate UMI assignment. The use of UMIs is even more important for long read than for short-read sequencing. Chimeric cDNA generated during PCR amplification is not an issue in short-read sequencing when only one extremity of the cDNA is sequenced. Conversely, PCR artifacts can severely affect the quality of long-read sequencing data. UMIs allow elimination of most of those PCR artifacts (Supplementary Fig. 4a). This will be particularly important for the identification of rare chimeric transcripts in single-cell studies of tumor heterogeneity.

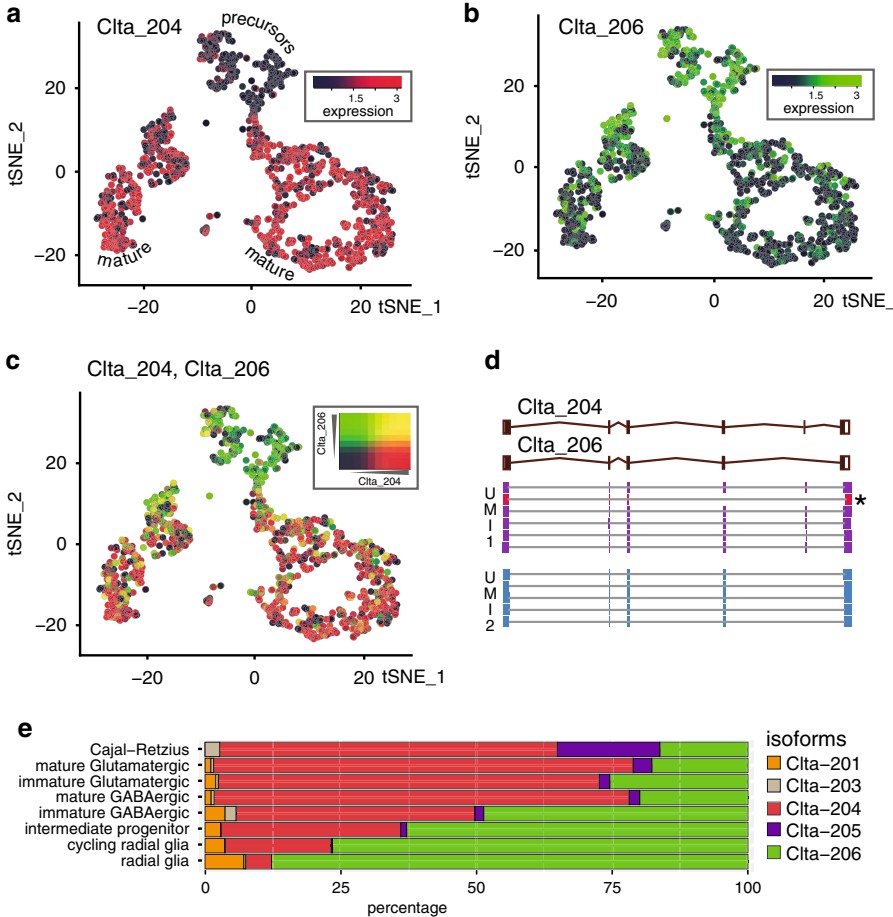

**Fig. 3 Nanopore scRNA-seq reveals transcript isoform diversity. a–c** Clta isoform expression switch during neuronal maturation visualized on the t-SNE plot of Fig. 2d. **a** Clta-204 (ENSMUST00000107849.9), **b** Clta-206 (ENSMUST00000170241.7), **c** Overlay Clta-204/Clta- 206. **d** Principal Clta splice variants and supporting genome-aligned reads for two UMIs. A read not consistent with the UMI consensus (likely PCR artifact) is labeled with an "*". **e** Relative expression of known Ensembl Clta transcript isoforms in different cell types of the brain (clusters of Fig. 2d).

In conclusion, ScNaUmi-seq can be easily plugged into standard single-cell sequencing workflows and should facilitate high throughput single-cell studies on RNA splicing, editing, and imprinting. We anticipate its usefulness in many biological and medical applications, from cell biology and development to clinical analyses of tumor heterogeneity.

## Methods

**Mouse brain dissociation**. A combined hippocampus, cortex, and ventricular zone pair from an E18 C57BL/6 mouse was obtained from BrainBits LLC (Leicestershire, UK). A single-cell suspension was prepared following the 10x Genomics protocol for "Dissociation of Mouse Embryonic Neural Tissue". Briefly, tissues were incubated with 2 mg/ml Papain in calcium free Hibernate E medium (BrainBits) for 20 min at 37 °C, rinsed with Hibernate-E/B27/GlutaMAX (HEB) medium, triturated and filtered through a 40 μm FlowMi cell strainer (Sigma-Aldrich). Cell concentration and viability were determined with a Countess® II automated cell counter (Life Technologies).

**Single-cell cDNA library preparation**. The E18 mouse brain single-cell suspension was converted into a barcoded scRNA-seq library with the 10x Genomics Chromium Single Cell 3′ Library, Gel Bead & Multiplex Kit and Chip Kit (v2), aiming for 1400 cells following the manufacturer's instructions with the following modifications: To obtain batches with different cell numbers, we split the emulsion before reverse transcription into two aliquots (85.7% and 14.3%; targeting 1200 and 200 cells). We extended the PCR elongation time during the initial PCR amplification of the cDNA from the manufacturer recommended 1 min to 3 min to minimize preferential amplification of small cDNAs. Half of the amplified cDNA was used for short-read sequencing library preparation following the 10x Genomics protocol and sequenced on an Illumina Nextseq 500 sequencer (26bases + 57bases). We generated 43 M and 70 M Illumina reads for the samples targeting 200 and 1200 cells respectively and mapped them to the mouse genome (build mm10) with the 10x Genomics Cell Ranger software (v2.0.0).

For each PromethION flow cell (Oxford Nanopore) we re-amplified 2–10 ng of the 10x Genomics PCR product for eight cycles with the primers NNNAAGCA GTGGTATCAACGCAGAGTACAT and NNNCTACACGACGCTCTTCCG ATCT (Integrated DNA Technologies, IDT). The Ns at the 5′ end of the primers avoid the preferential generation of reverse Nanopore reads (>85%) we observed without those random nucleotides. Amplified cDNA was purified with 0.65x SPRISelect (alternatively: 0.45x SPRISelect for depletion of cDNA smaller than 1 kb) and Nanopore sequencing libraries were prepared with the Oxford Nanopore LSK-109 kit (PCR free) following the manufacturer's instructions. For the libraries targeting 200 and 1200 cells, we generated 32 M reads and 322 M reads, respectively.

*Optional steps for the depletion of cDNA lacking a terminal poly(A)/poly(T)*. Amplified single-cell cDNA contains to a variable extend (30–50%) cDNA that lacks poly(A) and poly(T) sequences. For the study presented here, we did not deplete those cDNAs. Such cDNA can be depleted after a PCR of 2–10 ng of the 10x Genomics PCR product for 5 cycles with 5′-NNNAAGCAGTGGTATC AACGCAGAGTACAT-3′ and 5′ Biotine-AAAAACTACACGACGCTCTTC CGATCT 3′. After 0.55x SPRIselect purification to remove excess biotinylated primers, biotinylated cDNA (in 40 μl EB) is bound to 15 μl 1x SSPE washed Dynabeads™ M-270 Streptavidin beads (Thermo) resuspended in 10 μl 5x SSPE for 15 min at room temperature on a shaker. After two washes with 100 μl 1x SSPE and one wash with 100 μl EB, the beads are suspended in 100 μl 1x PCR mix and amplified for 6–9 cycles with the primers NNNAAGCAGTGGTATCAACGCA-GAGTACAT and NNNCTACACGACGCTCTTCCGATCT to generate enough material for Nanopore sequencing library preparation.

All PCR amplifications for Nanopore library preparations were done with Kapa Hifi Hotstart polymerase (Roche Sequencing Solutions): initial denaturation, 3 min at 95 °C; cycles: 98 °C for 30 s, 64 °C for 30 s, 72 °C for 5 min; final elongation: 72 °C for 10 min, primer concentration was 1 μM.

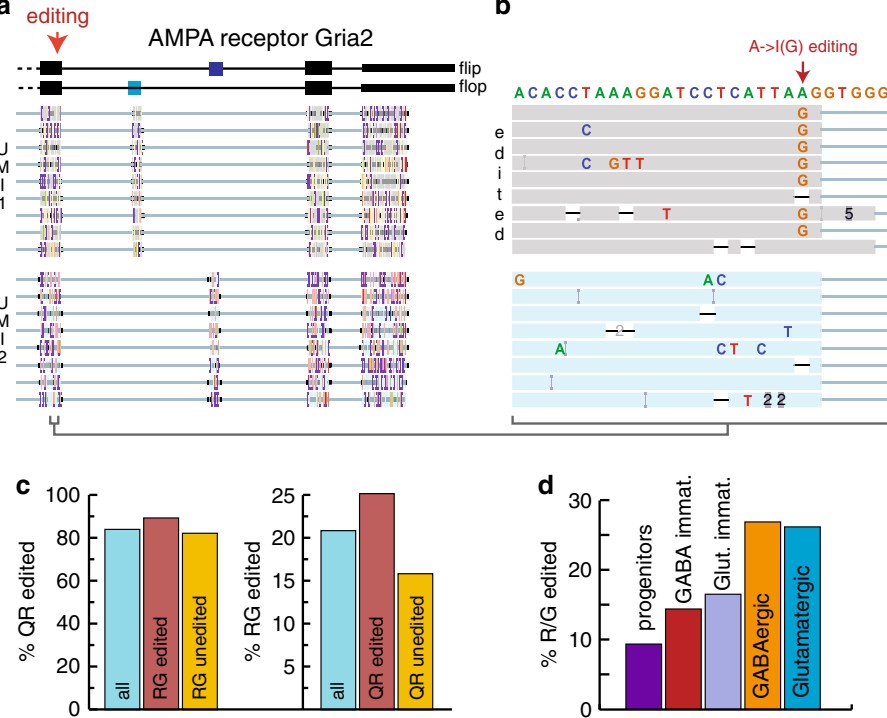

**Fig. 4 Nanopore scRNA-seq reveals sequence diversity. a** Principal splice variants of Gria2. The 3′ end of genome aligned reads for two UMIs are shown (Integrated Genome Viewer, UCSC Santa Cruz). **b** Zoom onto the R/G A- > I editing site for the reads shown in **e**. **c** Q/R and R/G editing. Editing rates at the Q/R site are 89.1 and 82% for RNAs edited or unedited at the R/G site respectively. R/G site editing is 25.1% and 15.8% for Q/R site edited and unedited RNAs respectively. **d** Extend of A− > I editing of the R/G site increases during neuronal maturation. Progenitors, 9.3%, clusters 1,2,5 in Fig. 2d; GABA immature, 14.3%, cluster 6; Glutamatergic immature, 16.4%, cluster 3; GABA mature, 26.7%, cluster 7; Glutamatergic mature, 26%, cluster 4.

**Mapping of Nanopore reads**. Nanopore reads were aligned to the *Mus musculus* Genome (mm10) with minimap2 v2.17 in spliced alignment mode (command: "minimap2 -ax splice -uf -MD -sam-hit-only -junc-bed"). The splice junction bed file was generated from the Gencode vM18 GTF using paftools.js, a companion script of minimap2. For reads matching known genes, the gene name was added to the corresponding SAM record (SAM Tag: GE) using the Sicelore (ScNaUmi-seq companion java toolkit) *AddGeneNameTag* method. Before cellBC and UMI assignment, SAM records were annotated with their Nanopore read sequence (SAM Tags: US) and read qualities (SAM tag: UQ) using the Sicelore *AddBam-ReadSequenceTag* method.

**CellBC and UMI assignment to Nanopore reads**. Our Java software performs the following analysis steps (Supplementary Fig. 2a):

1. *Parsing of Illumina data:* To retrieve accurate cellBC and UMI information, BAM files with the cellBC and UMI assigned Illumina data generated by the 10x Genomics Cell Ranger software were parsed. For each gene, we extracted the cellBC from the Illumina data and identified the UMIs found for each gene/cellBC combination. We also associated genomic regions (window size 500 nt.) with cellBCs and UMIs to account for reads that match outside of annotated genes. The parsed Illumina data were stored in nested Hash tables as serialized Java objects.

2. *Search for poly(T) tail:* Our software searches for a poly(T) and a poly(A) sequence (default 85% A or T, ≥20 nt.) within 100 nucleotides from the 5′ or 3′ end of the read, respectively. Reads without poly (A or T) and reads with a poly(A or T) on both ends are not further analyzed.

3. *Search for 3′ adapter sequence:* The cellBC and the UMI are located between an adapter sequence (10x Genomics 3′ PCR priming site) and the poly(T) of the reverse transcription primer (Supplementary Fig. 2b). To define the position of the cellBC we searched for the adapter sequence between the extremity of the read and the poly(A/T) sequence identified in the previous step using sliding window Needleman Wunsch alignments. The position with the best adapter match (least mismatches) was used. We found that searching for just the ten 3′ nucleotides of the adapter with 3 allowed mismatches (adapter found in 90.4% of the reads with poly(A)) more efficient than a search for a 20 nucleotide adapter sequence with 6 allowed mismatches (79.9% adapter found in poly(A) reads). Possible reasons for this are (i) The adapter 5′ end is very close to the extremity of the read and read quality might be lower there. (ii) Intrinsic error rate of Nanopore sequencing might be higher for the 5′ of the adapter sequence.

4. *Search for internal adapter and poly(A):* To flag reads corresponding to chimeric cDNA generated during library preparation, we searched for internal adapter sequences in proximity of a A- or T-rich sequence and flagged those reads as chimeric reads in the output file. In our dataset we found internal adapters in 3.5% of the reads.

5. *Search for cellBCs:* Cell barcodes in high accuracy Illumina reads are typically assigned by grouping reads that differ by not more than one position (edit distance: ED = 1). Indels are typically not considered. In consequence just 48 possible permutations of the 16 nt. cellBC reads need to be analyzed and assignment of reads to a barcode is highly reliable. Nanopore reads still have a mean error rate of about 5–10% (substitutions and indels). In consequence higher edit distances need to be examined and indels must be considered. This implies the generation and analysis of $\sum_{i=0}^{ED} 128^i$ barcode sequence permutations (2,113,664 for ED = 3) for each read. A 50 million read PromethION sequencing run would require the generation and analysis of about $10^{14}$ barcode sequences for ED = 3. This is clearly not feasible using reasonable sized compute clusters and standard bioinformatics approaches where sequences are typically treated as text. To solve this computational bottleneck, we encoded the barcode sequences using 2 bits per base (A: 00, G:01, T:10, C:11) which allows encoding of the entire 16 nucleotide cellBC into just one integer. This bitwise encoding allows performing substitutions, insertions, and deletions using highly efficient bitwise operations that require just one CPU cycle. Encoding cellBC and UMIs into integers also tremendously accelerates the search for matching Illumina cellBC or UMIs, since searching for a matching integer is much faster than searching for matching strings. An additional challenge for accurate barcode assignment is that 10x Genomics cellBCs are randomly selected out of a pool of 750,000 barcode sequences. The used barcode sequences are not known in advance. Clustering the Nanopore barcode reads correctly without a priori knowledge of the used barcodes is rather error prone, since two reads that each have e.g., two sequencing errors in the cellBC, can differ in up to four positions when compared to each other. To improve the accuracy of cell barcode assignment, we used the cellBC sequences defined by Illumina sequencing of the same libraries to guide the cellBC identification in Nanopore reads. For each genome aligned Nanopore read, we extracted the sequence (16 nt. barcode plus 4 nt. to allow insertion errors) just downstream of the adapter (position corrected for indels in adapter) identified in the previous step. We also extracted the barcode sequences for the preceding and following position to account for eventual terminal indels in the adapter read. We then compared the extracted

Nanopore barcode sequences and all possible permutation up to a defined edit distance with: (i) The cellBCs identified by Illumina sequencing for the same gene if the Nanopore read matches a known gene in the Illumina dataset; (ii) Illumina cellBCs sequences for the genomic region (500 nt. window) if the Nanopore read matches an unannotated genomic region or if the gene name was not found in the Illumina data. Since the Nanopore barcodes are only compared with Illumina barcodes found for the same gene or genomic region, the complexity of the search set is reduced. Barcode matches, eventual second-best barcode matches and their quality (edit distance) were recorded in the output BAM file. The maximal ED is dynamically selected (ED limit 1–4) and depends on the number of cellBCs found for the same gene or genomic region in the Illumina data (the complexity of the search set). The software allows to define the maximal false assignment percentage as a parameter. We used computer simulations of collision frequencies to define the complexity of the search set (number of Illumina barcodes) allowed for different edit distances and maximal false assignment percentages. Simulation data are supplied as an XML file and can be easily adapted. Optionally the software also allows the definition of fixed edit distance limits.

6. *Search for UMIs:* To identify the ten nucleotide UMI, 14 nucleotides (to allow for insertions in the read) following the end of the cellBC sequence (position corrected for barcode indels) were extracted. For UMI assignment, we used the same strategy as for cellBC assignment. We searched for matching UMIs in Illumina data for the same cellBC and either genomic region or gene identified in the previous step. This means the complexity of the search set corresponds to the copy number of a given gene in just one cell. The maximal allowed edit distance which is dynamically adjusted depending on the complexity of the search set (detailed above).

7. *Determination of cellBC and UMI assignment accuracy:* To evaluate the accuracy of cellBC and UMI assignment, we first scanned each Nanopore SAM record for a matching Illumina cellBC or UMI. We then repeated the scan where we replaced each cellBC or UMI sequence extracted from the Nanopore read by a random sequence and mutated this random sequence allowing the same number of mutations (edit distance) that was used for the same SAM record in the previous scan and searched for matching cellBC or UMIs in the Illumina dataset. The ratio between the number of reads that were assigned to both a cellBC and UMI after and before replacement of the cellBC or the UMI against a random sequence is the false assignment probability.

The cellBC assignment accuracy is particularly high, since after an incorrect cellBC assignment, the UMI is compared with the UMIs associated with this wrong cellBC in the Illumina data. In consequence, most reads with falsely assigned cellBC are subsequently eliminated during UMI assignment. The false barcode assignment rate in cellBC and UMI assigned reads is thus the product of the false cellBC and the false UMI discovery rate. With the default cellBC (5% false assignment) and UMI scanning (2% false assignment) parameters we used, we obtained an effective cellBC assignment accuracy of 99.8%, which is close to the value expected with those parameters. In a second approach to assess the accuracy of cellBC and UMI assignment we guided the barcode and UMI assignment to the Nanopore reads with Illumina short-read sequencing data from an independent mouse brain single-cell sequencing experiment. The unrelated Illumina dataset was from an 885 cell P18 mouse brain sample (GEO accession number GSM4224249, 219k reads/cell, 88.1% sequencing saturation, 6550 UMIs/cell, 2739 genes/cell).

While in the first approach (replacement against random sequences), the maximal edit distance tested for each read was limited to the edit distance for which a match was found (if any) with the non-random sequences, in the second approach, the cellBC and UMI search was allowed to proceed up to the maximal allowed edit distance. E.g. when for a given Nanopore read a cellBC match was found with the correct Illumina dataset at ED = 0, a cellBC match at a higher edit distance in the unrelated Illumina dataset was considered a match.

8. *Maximal possible Barcode and UMI assignment efficiency with 10xGenomics data:* Since our software scans for matches of the Nanopore cell barcodes with barcodes associated with cells (the relevant barcodes), barcodes associated with empty drops are ignored. The maximal possible barcode discovery rate in Nanopore reads corresponds to the percentage of reads associated with cells in the Illumina dataset: 83.7% for the 190 cell and 85.1% for the 951 cell sample, respectively.

The maximal UMI discovery rate for Nanopore reads depends on the Illumina sequencing depth of the same sample and corresponds to the sequencing saturation computed by the 10xGenomics Cell Ranger software after Illumina sequencing. The sequencing saturation is the probability that a matching UMI is found in the Illumina dataset for a given read. For the 190 cell and 951 cell samples, the sequencing saturations were 90.5% and 74.8% respectively. Efficiencies of barcode assignment are given as percentages of cell associated barcodes. UMI assignment efficiencies were corrected for the sequencing saturation.

9. *Compatibility of the software:* The software is compatible with the 10x Genomics workflow v2 and the recent upgrade (v3) which uses 12nt UMIs.

It can also be used for cellBC and UMI assignment of long-read single-cell data generated with other single-cell isolation systems with the following limitations: The cDNA needs to have a 3′ adapter followed by a cellBC, an UMI and a poly(A). CellBC and UMI length, adapter sequences as well as the search stringency for poly(A/T), adapter, cellBC, and UMI can be configured accordingly.

**Definition of cDNA consensus sequences for each UMI.** Potentially chimeric reads (terminal Soft/Hard-clipping of >150 nt; 1.93% and 3.95% for the 190 and 951 dataset respectively) and reads with low-quality genome alignments (minimap2 mapping quality values = 0) were filtered out. SAM records for each cell and gene were grouped by UMI. The cDNA sequence, between TSO end (TE SAM tag) and poly(A) start (PE SAM tag), was extracted for consensus sequence computation using the Sicelore *ComputeConsensus* method. Depending on the number of available reads for the UMI the following sequence was assigned as the consensus sequence for the UMI: (i) just one read, the cDNA sequence of the read was assigned to the UMI; (ii) two reads, the cDNA sequence of the best mapping read (lowest "de" minimap2 SAM record tag value) was defined as the consensus sequence; (iii) More than 2 reads, a consensus sequence of all cDNA sequences for the UMI was generated after poa[13] multiple alignment and polished with racon[14] using all cDNA sequences for the UMI.

**Assignment of Gencode transcript isoforms.** Consensus cDNA sequences for all UMIs were aligned to the *Mus musculus* Genome (mm10) with minimap2 v2.17 in spliced alignment mode. SAM records matching known genes were analyzed for matching Gencode vM18 transcript isoforms (same exon makeup). To assign a UMI to a Gencode transcript we required a full match between the UMI and the Gencode transcript exonic structures. We authorized a two-base margin of added or lacking sequences at exon boundaries, to allow for indels at exon junctions and imprecise mapping by minimap2. Following this strategy (Sicelore *IsoformMatrix* method), we assigned 63.6% of the UMIs to a known Gencode transcript isoform and generated gene-level (median UMIs/cell = 6047) and isoform-level (median UMIs/cell = 3795) count matrices used for the Nanopore/Illumina gene count and UMI count per cell correlations (Fig. 2a, b) and for the transcripts isoforms t-SNE (Fig. 2e).

**Single-cell gene expression quantification and determination of major cell types.** Raw gene expression matrices generated by Cell Ranger were processed using R/Bioconductor (version 3.5.2) and the Seurat R package (version 3.1.4). A total of 190 cells and 951 cells were detected with default Cell Ranger cutoffs for the two replicates. Cells with over 95% dropouts were removed. From the 186 and 935 remaining cells (hereafter called 1121 cells dataset), gene expression matrices were cell level scaled to 10.000 and log-normalized. The top 2000 highly variable genes were selected based on the variance-stabilizing transformation method and used for Principal Component Analysis. Due to differences in sequencing depth of both replicates, data were integrated using the Seurat CCA method. The first 11 aligned canonical correlations were used to define the integrated sub space for clustering and t-SNE visualization of the 1121 remaining cells. Clusters in the t-SNE plot were assigned to known cell types using canonical marker genes (Supplementary Fig. 8a, Supplementary Data 1). Using Seurat multi-modal capabilities, we integrated Illumina and Nanopore gene-level and isoform-level datasets allowing direct comparison of gene and isoform expression in individual cells.

**Identification of novel transcript isoforms.** UMIs of the 1121 cells dataset were used for the identification of novel transcripts isoforms using the Sicelore *CollapseModel* method. cDNA consensus sequences for UMIs with an exon structure not supported by Gencode (at least one splice junction different when compared with annotated Gencode transcripts, see Supplementary Fig. 6, Supplementary Note) were first grouped by gene and sequences with identical exon structure were used to define potential novel transcript isoforms. Novel isoforms supported by less than five UMIs were discarded. Isoforms with identical exon layout that differ in SNVs or 5′ or 3′ ends were considered as identical isoforms. Novel isoforms were classified as suggested by Tardaguila et al.[15]: (i) combination of known splice junctions, only composed of exon-exon junctions found in Gencode transcripts; (ii) combination of known splice sites, individual donor and acceptor sites are known, but the resulting splice junction is novel; (iii) at least one donor or acceptor site is not found in Gencode transcripts.

We next filtered novel transcripts isoforms requiring: (i) all exon-exon junctions either described in Gencode or confirmed in an E18 cortex/ midhindbrain Illumina short-read dataset (GEO accession GSE69711); (ii) a 5′ end located within 50 nucleotides of a known transcription start site identified by CAGE (FANTOM5 mm9 reference UCSC liftover to mm10, https://fantom.gsc.riken.jp/5/datafiles/latest/extra/CAGE_peaks/); (iii) a 3′ end within 50 nucleotides of a polyadenylation site (GENCODE vM24 PolyA feature annotation, https://www.gencodegenes.org/mouse/).

**Correlation of gene or isoform expression between replicates.** To analyze expression correlations between clusters, Illumina gene-level and Nanopore

gene- and isoform-level data were downsampled (R package DropletUtils[16] downsampleMatrix method) to the median UMIs/cell of the 1121 cell Nanopore transcript isoform-level dataset (3795 UMIs/cell). We then grouped for each replicate cells for each cluster (see Fig. 2d, e), and used the mean expression of each gene or isoform in the clusters to produce a Pearson correlation matrix (R cor function). Heatmaps in Supplementary Fig. 7 were generated after cluster agglomeration with the Ward method (pheatmap package).

**Gria2 data analysis**. 9593 reads (2105 UMIs) corresponding to Gria2 (mm10: chr3:80,682,936-80,804,791) were extracted from the 951 and the 190 cell dataset. A consensus sequence for each molecule was computed and re-mapped to the mm10 genome for SNP calling using the Sicelore SNPMatrix method. Gria2 mRNAs are huge (> 6 kB) and inefficiently converted into full-length cDNA in the 10x Genomics workflow. This is likely due to: (i) some RNA degradation within the droplet between cell lysis and reverse transcription. (ii) internal reverse tran-scription priming at A-rich sites within the cDNA leading to cDNAs that cover only part of the transcripts. In consequence, we noticed 3′ bias and fragmented coverage for certain long transcripts such as Gria2 where a total of 456 cDNA molecules (UMIs) had the R/G-editing-and 233 had both the R/G and the Q/R-editing-site. Further optimization of the 10x Genomics workflow should allow more efficient full-length capture of long mRNAs.

**Reporting summary**. Further information on research design is available in the Nature Research Reporting Summary linked to this article.

## Data availability
All relevant data have been deposited in Gene Expression Omnibus under accession number GSE130708.

## Code availability
Source data are provided with this paper. All custom software used is available on Github https://github.com/ucagenomix/sicelore.

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

## Acknowledgements
This project was funded by grants from the Institut National contre le Cancer (PLBIO2018-156), the Conseil Départemental des Alpes Maritimes (2016-294DGADSH-CV), FRM (DEQ20180339158), the Inserm Cross-cutting Scientific Program HuDeCA 2018, the National Infrastructure France Génomique (Commissariat aux Grands Investissements, ANR-10-INBS-09-03, ANR-10-INBS-09-02). This publication is part of the Human Cell Atlas - https://www.humancellatlas.org/publications.

## Author contributions
Conception and design: R.W., P.B.; Experimental work: V.M., R.W.; Data process, curation and visualization: K.L., R.W.; Analysis and interpretation: K.L., R.W., P.B.; Drafting of the manuscript: R.W.; Manuscript review and edition: all authors. Funding and supervision: P.B.

## Competing interests
The authors declare no competing interests.
