## [Peer Review File · Nature Communications]

Reviewers' Comments:

Reviewer #1:

Remarks to the Author:

The authors have addressed my concerns on the description of the transcript novelty. It was nice to see that novel transcripts are detected at least by 2 cells.

However, I would strongly suggest they use SQANTI categories to describe the novelty as this is the standard in the field and most of their definitions of novel transcript classes correspond to an already defined SQANTI category.

The authors should be sure none of the novel transcripts have non-canonical splice sites. I do not expect this to be a major problem because of the 2 cells filter, but just in case.

Reviewer #1 (Remarks to the Author):

The authors have addressed my concerns on the description of the transcript novelty. It was nice to see that novel transcripts are detected at least by 2 cells.

We are happy that Reviewer #1 has been satisfied by the new description of the transcript novelty.

However, I would strongly suggest they use SQANTI categories to describe the novelty as this is the standard in the field and most of their definitions of novel transcript classes correspond to an already defined SQANTI category.

Regarding SQANTI, we have added a reference to cite the original work by Tardaguila et al on page 18:

"Novel isoforms were classified as suggested by Tardaguila et al 15: (i) combination of known splice junctions, only composed of exon-exon junctions found in Gencode transcripts; (ii) combination of known splice sites, individual donor and acceptor sites are known, but the resulting splice junction is novel; (iii) at least one donor or acceptor site is not found in Gencode transcripts."

Our classification of novel transcripts was actually the same as the SQANTI classification. We initially used slightly different names for some categories. We changed the names of the categories in Supplementary Fig. 6 and in the text. We now use SQANTI names for the categories.

The authors should be sure none of the novel transcripts have non-canonical splice sites. I do not expect this to be a major problem because of the 2 cells filter, but just in case.

This point is only relevant for the 1,577 transcripts with "at least one novel splice site", since the "known junctions" and "known splice sites" categories are already described in Gencode.

We found 56 non canonical sites out of these 1,577 transcripts. First, they were all detected by Nanopore AND by Illumina sequencing. Second, these sequences were present in at least 5 different cells and were detected with at least 5 distinct UMIs. For these 2 reasons, we considered these sequences as valid.